# Essential role of TMPRSS2 in SARS-CoV-2 infection in murine airways

Naoko Iwata-Yoshikawa[1,7], Masatoshi Kakizaki[2,7], Nozomi Shiwa-Sudo[1], Takashi Okura[2], Maino Tahara[2], Shuetsu Fukushi[3], Ken Maeda[4], Miyuki Kawase[2], Hideki Asanuma[5], Yuriko Tomita[5], Ikuyo Takayama[5], Shutoku Matsuyama[5], Kazuya Shirato[2], Tadaki Suzuki[1], Noriyo Nagata [1] ✉ & Makoto Takeda [2,6] ✉

In cultured cells, SARS-CoV-2 infects cells via multiple pathways using different host proteases. Recent studies have shown that the furin and TMPRSS2 (furin/TMPRSS2)-dependent pathway plays a minor role in infection of the Omicron variant. Here, we confirm that Omicron uses the furin/TMPRSS2-dependent pathway inefficiently and enters cells mainly using the cathepsin-dependent endocytosis pathway in TMPRSS2-expressing VeroE6/TMPRSS2 and Calu-3 cells. This is the case despite efficient cleavage of the spike protein of Omicron. However, in the airways of TMPRSS2-knockout mice, Omicron infection is significantly reduced. We furthermore show that propagation of the mouse-adapted SARS-CoV-2 QHmusX strain and human clinical isolates of Beta and Gamma is reduced in TMPRSS2-knockout mice. Therefore, the Omicron variant isn't an exception in using TMPRSS2 in vivo, and analysis with TMPRSS2-knockout mice is important when evaluating SARS-CoV-2 variants. In conclusion, this study shows that TMPRSS2 is critically important for SARS-CoV-2 infection of murine airways, including the Omicron variant.

At the end of 2019, an emerging infectious respiratory viral disease (coronavirus disease 2019, COVID-19) occurred in Wuhan, China and spread worldwide within a few months[1,2]. The World Health Organization (WHO) declared a global pandemic on March 11, 2020. This unprecedented pandemic is caused by severe acute respiratory syndrome coronavirus 2 (SARS-CoV-2) of the *Sarbecovirus* subgenus of the *Betacoronavirus* genus and is currently ongoing. The spike (S) glycoprotein of SARS-CoV-2 is an essential determinant of host range and cell susceptibility and is also a significant target of antibody responses induced by infection and vaccination. The S protein consists of S1, which contains a receptor-binding domain (RBD) that binds to angiotensin-converting enzyme 2 (ACE2) and S2, which mediates membrane fusion[3,4]. Host protein-mediated S protein cleavage at the S1/S2 and S2' sites is necessary to induce membrane fusion. Notably,

the S1/S2 site of SARS-CoV-2 contains a suboptimal furin cleavage motif (FCM) (682-RRAR ↓ S-686; arrows indicate cleavage sites) that is not found in other sarbecoviruses. S1/S2 cleavage promotes binding of the S protein to ACE2[5,6] and cleavage of the S2' site[7], which occurs after binding to ACE2[5,8]. TMPRSS2, a type II transmembrane serine protease, is a significant protease that mediates S2' cleavage and supports viral entry from the plasma membrane (the early entry pathway)[8–10]. SARS-CoV-2 also enters cells via endocytosis (the late entry pathway)[11] and uses lysosomal cathepsin L to cleave the S protein, supporting viral entry in this cellular compartment[11]. FCM at the S1/S2 site is not required for SARS-CoV-2 to enter cells via the late entry pathway and may, in fact, be detrimental in exploiting this pathway[8,12].

In spring 2020, a SARS-CoV-2 variant harbouring a D614G mutation in the S protein emerged[13], and this was followed by the

[1]Department of Pathology, National Institute of Infectious Diseases, Tokyo, Japan. [2]Department of Virology III, National Institute of Infectious Diseases, Tokyo, Japan. [3]Department of Virology I, National Institute of Infectious Diseases, Tokyo, Japan. [4]Department of Veterinary Science, National Institute of Infectious Diseases, Tokyo, Japan. [5]Center for Influenza and Respiratory Virus Research, National Institute of Infectious Diseases, Tokyo, Japan. [6]Department of Microbiology, Graduate School of Medicine and Faculty of Medicine, The University of Tokyo, Tokyo, Japan. [7]These authors contributed equally: Naoko Iwata-Yoshikawa, Masatoshi Kakizaki. ✉e-mail: nnagata@niid.go.jp; mtakeda@m.u-tokyo.ac.jp

emergence of a variety of variants with the D614G mutation and a different set of mutations. Specific variants were more contagious than other variants, further increasing the threat of the SARS-CoV-2 pandemic. Such variants were designated variants of concern (VOCs), indicating that their epidemiological, immunological, or pathogenic properties represented a potential threat. The WHO assigned each variant a Greek letter. The Alpha (Pango lineage B.1.1.7), Beta (B.1.351) and Gamma (P.1) VOCs were first detected in the United Kingdom[14], South Africa[15] and Brazil[16], respectively, in late 2020 and then spread worldwide. Subsequently, the Delta VOC (B.1.617.2a) was detected in India in March 2021[17] and spread globally, then in October 2021, a new VOC (Omicron, B.1.1.529) first detected in South Africa began to spread[18].

Each VOC has characteristic mutations, among which the P681R mutation in the S1/S2 site in the Delta VOC has been shown to increase the cleavability of the S protein[19] and fusogenicity, which are associated with high virulence in animal models[20]. In addition, structural bioinformatics approaches reveal that the Alpha variant allows tighter binding of the S protein with furin because of the P681H mutation in the S1/S2 site[21]. However, biochemical and virological analyses show that the P681H mutation has little impact on furin cleavage and virus infectivity[22]. Surprisingly, the Omicron VOC harbours -30 mutations in the S protein, of which P681H is one of the mutations[23]. The Omicron VOC has another mutation (N679K) at the S1/S2 site[23]. These mutations were expected to increase S protein cleavage by furin. However, recently published papers reported that the Omicron S protein is less cleavable by furin[24–27] and the Omicron VOC uses mainly the endocytosis pathway rather than the furin/TMPRSS2-mediated pathway, unlike the other VOCs[24,27–30]. In this study, we analysed various SARS-CoV-2 strains, including the VOCs, in vitro to understand the changes in S protein cleavability and the infection mode of virus entry. We also analysed the replication capacity of these SARS-CoV-2 strains in normal mice and TMPRSS2-knockout (KO) mice[31] to show the significance of TMPRSS2 in the replication and pathogenicity of each SARS-CoV-2 strain in vivo. Our data showed that TMPRSS2-mediated infection is essential for the replication and pathogenicity of SARS-CoV-2 in murine airways and that the Omicron VOC is not an exception in using TMPRSS2 in vivo.

## Results

### Cleavability of S protein

VeroE6/TMPRSS2 cells, which express TMPRSS2, are highly susceptible to SARS-CoV-2 infection[9]. We showed that all SARS-CoV-2 isolates replicated efficiently in VeroE6/TMPRSS2 cells (Fig. 1a). However, the replication efficiency of the Omicron variant was significantly lower than that of the other strains (Fig. 1a). Western blot analysis of the S protein showed a 1.5- to 2.8-fold increase in cleavage of S protein in the Alpha and Omicron variants compared with the Wuhan-type (WK521) strain, and the Beta and Gamma variants (Fig. 1b, Supplementary Fig. 1a–d). The cleavage of S protein was more markedly increased in the Delta and Kappa variants (Fig. 1b). Expression plasmids were also used to evaluate the cleavability and fusion ability of the S protein. The S proteins of Wuhan, Alpha, Beta, Gamma, Delta and Omicron were expressed in VeroE6/TMPRSS2 cells. The Alpha S protein produced larger syncytia than the Wuhan-type S protein (Fig. 1c). The syncytia induced by Delta S protein were even more extensive than those induced by the Alpha S protein (Fig. 1c). By contrast, the size of syncytia produced by the Omicron S protein was significantly smaller than the sizes of those produced by the S proteins of other variants and the Wuhan-type strain (Fig. 1c).

### Entry pathway in VeroE6/TMPRSS2 cells

VeroE6 cells express high levels of cathepsin L[32]. Therefore, SARS-CoV-2 efficiently utilises the early entry and endocytosis pathways in VeroE6/TMPRSS2 cells. Monolayers of VeroE6/TMPRSS2 cells were

infected with various SARS-CoV-2 strains and cultured using an agarose-containing medium in the presence or absence of (2 S,3 S)-trans-epoxysuccinyl-L-leucylamindo-3-methylbutane ethyl ester (EST), which inhibits the cathepsin L-mediated endocytosis pathway. EST reduced the number of plaques caused by the Wuhan-type strain and the Beta, Gamma and Omicron variants (Fig. 1d, e, Supplementary Fig. 1e). These data indicated that a significant portion of these viral infections in VeroE6/TMPRSS2 cells were dependent on the endocytosis pathway. By contrast, the number of plaques caused by the Alpha, Delta and Kappa variants was only slightly reduced or not affected by the presence of EST (Fig. 1d, e, Supplementary Fig. 1e). Thus, the furin/TMPRSS2-dependent early entry pathway can be used almost exclusively by these viruses to achieve infection.

We also performed a similar assay using normal Vero cells, in which only the cathepsin L-mediated endocytosis pathway is available. The Wuhan-type strain and the Omicron variant produced an equivalent number of plaques in Vero cells as in VeroE6/TMPRSS2 cells (Supplementary Fig. 1f), indicating that these viruses mainly utilise the cathepsin L-mediated endocytosis pathway in Vero and VeroE6/TMPRSS2 cells. By contrast, the numbers of plaques produced by the Alpha, Beta, Gamma, Delta and Kappa variants were reduced by 50% to 90% in Vero cells. These data indicated that the early entry pathway using TMPRSS2 is necessary for these viruses to enter cells efficiently (Supplementary Fig. 1f).

### Growth and entry pathway in Calu-3 cells

Calu-3 cells endogenously express a significant level of TMPRSS2, but cathepsin L is expressed at marginal levels[32]. We examined the replication kinetics of the Wuhan-type strain and the Alpha, Beta, Gamma, Delta, Kappa and Omicron variants in Calu-3 cells. Unlike in VeroE6/TMPRSS2 cells, replication of the Wuhan-type strain was severely restricted in Calu-3 cells (Fig. 2a), whereas all variants, except for Omicron, replicated efficiently in Calu-3 cells (Fig. 2a). Among the variants, the replication rate of Delta was the greatest, and the replication rate of Omicron was close to the detection limit (Fig. 2a). Calu-3 cells were also infected with SARS-CoV-2 in the presence or absence of protease inhibitors (EST or nafamostat) to investigate which entry pathway was preferentially utilised. At 24 h post-infection (p.i.), total RNA was purified from SARS-CoV-2-infected Calu-3 cells and real-time PCR was performed to quantify the number of viral RNAs. The Omicron variant showed a poor propagation capacity in Calu-3 cells, and a higher amount of virus [multiplicity of infection (MOI) = 1.0] was necessary to assess the infection. This low level of infection of Calu-3 cells with Omicron was not inhibited by nafamostat, while it was moderately inhibited by EST (Fig. 2b). Similarly, the Wuhan-type strain showed a low replication capacity in Calu-3 cells and both EST and nafamostat inhibited this low level of infection (Fig. 2b). For the other variants, nafamostat, but not EST, inhibited the infections efficiently (Fig. 2b). Thus, all variants, except for Omicron, preferentially utilised the TMPRSS2-dependent early entry pathway in Calu-3 cells. If the cathepsin-mediated endocytosis pathway can also be used for SARS-CoV-2 infection in Calu-3 cells, then a certain level of viral infection should be observed even when TMPRSS2 activity is inhibited by nafamostat. However, in Calu-3 cells, SARS-CoV-2 infection was almost completely inhibited by nafamostat alone (Fig. 2b). Given these results, we concluded that the cathepsin-mediated pathway cannot be a substitute for the TMPRSS2-mediated pathway in these cells.

### Infection of TMPRSS2-knockout mice with the mouse-adapted QHmusX strain

To examine the effect of TMPRSS2 expression during SARS-CoV-2 infection, we infected C57BL/6 wild-type (WT) and TMPRSS2-KO mice with a mouse-passaged SARS-CoV-2 strain, QHmusX ($1.9 \times 10^4$ TCID$_{50}$ in 25 μL). Our previous study[31] showed that these TMPRSS2-KO mice are highly tolerant to lethal dose infection with influenza virus strains

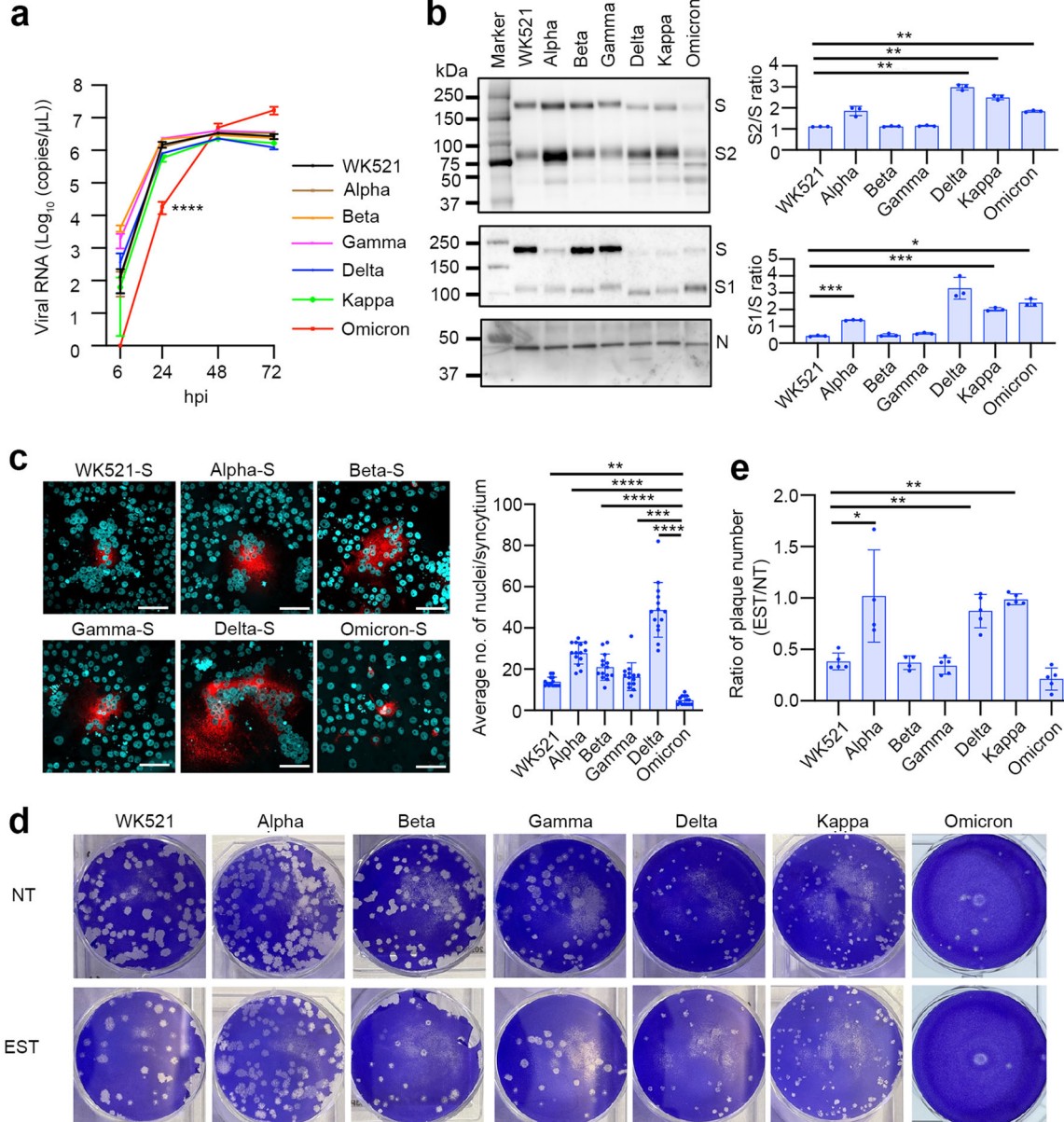

**Fig. 1 | The replication ability, spike protein cleavage, fusion capability and entry phenotype of SARS-CoV-2 in VeroE6/TMPRSS2 cells. a** Replication kinetics in cells infected with viruses at an MOI of 0.1. Viral RNA copy numbers in the culture supernatants were quantified at 6, 24, 48 and 72 h post-infection (p.i.). Error bars indicate the standard deviations (SD) of triplicate wells. Mean values ± SD are shown, and significant differences were determined with one-way ANOVA. Multiple comparisons between WK521 and other variants were adjusted with Dunn's multiple comparisons test, ****$P$ < 0.0001 at 24 hpi (vs Omicron). **b** S and N proteins detected in the culture supernatant of infected cells. SDS-PAGE and western blot analysis (left panel). Quantified S protein cleavage rate (bar graphs in the right panels). Data are representative of triplicate independent experiments. Mean values ± SD are shown, and statistical significance was determined with Brown–Forsythe and Welch ANOVA tests. Multiple comparisons between WK521 and other variants were adjusted with Dunnet's T3 multiple comparisons test, in S2 **$P$ = 0.0050 (vs Delta), **$P$ = 0.0064 (vs Kappa), **$P$ = 0.0037 (vs Omicron) and in S1 ***$P$ = 0.0002 (vs Alpha), ***$P$ = 0.0005 (vs Kappa) and *$P$ = 0.0116 (vs Omicron). **c** Syncytium formation in cells transfected with the S protein expression plasmid. S protein was detected by

an indirect immunofluorescent assay (red) and nuclei were detected by DAPI (blue) staining. The number of nuclei in 30 individual, randomly-selected syncytia were counted. Mean values ± SD are shown and significant differences were determined with one-way ANOVA. Multiple comparisons between Omicron-S and other variants-S were adjusted with Dunn's multiple comparison test, **$P$ = 0.0053 (vs WK521-S), ****$P$ < 0.0001 (vs Alpha), ****$P$ < 0.0001 (vs Beta), ***$P$ = 0.0002 (vs Gamma), and ****$P$ < 0.0001 (vs Delta). Scale bars indicate 50 μm. **d** Plaque formation in the absence (not treated: NT) or in the presence of EST. The same infectious titre of the virus was used for infection. **e** Reduction in the plaque number following EST treatment. The bar graph shows the ratio of the plaque number in EST-treated cells to that in untreated cells (NT). Error bars indicate the SD of quadruplicate or quintuplicate wells. Mean values ± SD are shown and significant differences were determined with the Kruskal–Wallis test. Multiple comparisons between WK521 and other variants were adjusted with Uncorrected Dunn's multiple comparison test, *$P$ = 0.0334 (vs Alpha), *$P$ = 0.0309 (vs Delta) and **$P$ = 0.0098 (vs Kappa). Wuhan-type strain (WK521), Alpha (QHN001), Beta (TY8-612), Gamma (TY5-501), Delta (TY11-927), Kappa (TY11-330) and Omicron (TY38-873) variants were used.

that use TMPRSS2 for hemagglutinin cleavage but are as susceptible as WT mice to the infection with H5N1 subtype influenza virus that uses furin, but not TMPRSS2, for hemagglutinin cleavage. We again confirmed these characteristics of the TMPRSS2-KO mice used in this study (Supplementary Fig. 2). Therefore, the TMPRSS2 KO condition in these mice specifically suppresses virus infection dependent on TMPRSS2 expression. After intranasal inoculation with the SARS-CoV-2 strain, QHmusX, WT mice showed transient body weight loss from 2 to

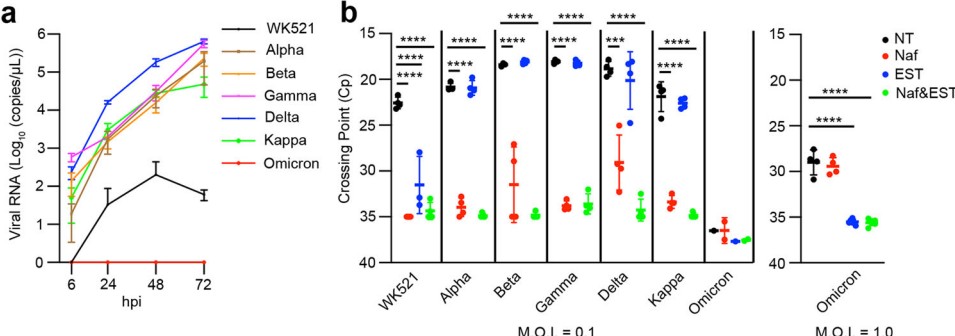

**Fig. 2 | Replication ability and entry phenotype of SARS-CoV-2 in Calu-3 cells.** **a** Replication kinetics in cells infected with viruses at an MOI of 0.1. Viral RNA copy numbers in the culture supernatants were quantified 6, 24, 48 and 72 h p.i. **b** Cells were infected with viruses at an MOI of 0.1 (left bar graph) or 1.0 (right bar graph) in the absence (NT) or presence of either nafamostat (Naf) or EST or both (Naf & EST). At 24 h p.i., viral RNAs in cells were detected by real-time RT-PCR. The x-axis shows the Cp values as an indicator of the viral RNA level. **a**, **b** Error bars indicate the standard deviations of triplicate wells. Mean values ± SD are shown. **b** Statistical significance was determined with two-way ANOVA. Multiple comparisons among different groups were adjusted with Tukey's multiple comparison tests, in WK521 ****$P < 0.0001$ (NT vs Naf), ****$P < 0.0001$ (NT vs EST) and ****$P < 0.0001$ (NT vs Naf&EST); in Alpha ****$P < 0.0001$ (NT vs Naf) and ****$P < 0.0001$ (NT vs Naf&EST); in Beta ****$P < 0.0001$ (NT vs Naf) and ****$P < 0.0001$ (NT vs Naf&EST); in Gamma ****$P < 0.0001$ (NT vs Naf) and ****$P < 0.0001$ (NT vs Naf&EST); in Delta ***$P = 0.0002$ (NT vs Naf) and ****$P < 0.0001$ (NT vs Naf&EST); in Kappa ****$P < 0.0001$ (NT vs Naf) and ****$P < 0.0001$ (NT vs Naf&EST); in Omicron ****$P < 0.0001$ (NT vs EST) and ****$P < 0.0001$ (NT vs Naf&EST). Wuhan-type strain (WK521), Alpha (QHN001), Beta (TY8-612), Gamma (TY5-501), Delta (TY11-927), Kappa (TY11-330) and Omicron (TY38-873) variants were used.

3 days, but this was not observed in TMPRSS2-KO mice ($n = 10$ or 8) (Supplementary Fig. 3a). Neutralising antibody titres were detected in the sera of 7/10 WT mice on day 9 p.i., while none of the KO mice showed seroconversion ($n = 10$ or 8) (Supplementary Fig. 3a). Viral titres in the lungs peaked on days 2 and 3 p.i. in WT mice but declined gradually by day 4 in KO mice ($n = 4$) (Supplementary Fig. 3a).

Histopathological and immunohistochemical analyses revealed that TMPRSS2 deficiency affects airway pathology after infection with the QHmusX strain. On day 1, strong viral antigen-positive cells were detected immunohistochemically in the bronchial epithelium of WT mice ($n = 4$) (Supplementary Fig. 3b, left panel) and by day 3, viral infection had expanded into the alveolar area ($n = 4$) (Supplementary Fig. 3b, second panel, red arrows). On day 5 p.i., progressive inflammatory infiltration of neutrophils and mononuclear cells was observed around the bronchioles and in the alveoli of WT mice, which subsided on day 9, along with infiltration of lymphocytes ($n = 4$) (Supplementary Fig. 3b). By contrast, no viral antigens or histopathological lesions were detected in the lungs of KO mice up to day 9 p.i. ($n = 10$ or 8) (Supplementary Fig. 3b).

Acute inflammatory chemokines and cytokines, including tumour necrosis factor (TNF)-α, IP-10/CXCK10, MIG/CXCL9, monocyte chemoattractant protein 1 (MCP-1)/CCL2 and macrophage inflammatory protein 1α (MIP-1α)/CCL3, were elevated in the early phase of infection in the lungs of WT mice. Then, interleukin (IL)−12 (p40), IL-6, keratinocyte-derived chemokine (KC/CXCL1), and granulocyte-macrophage colony-stimulating factor (GM-CSF) levels were increased at 2 or 3 days p.i. ($n = 4$) (Supplementary Fig. 3c, black bars). KO mice did not show any alteration in cytokine/chemokine levels in the lungs during the observation period (Supplementary Fig. 3c, red bars). These results indicated that deficiency of TMPRSS2 inhibited infection of the bronchi and alveoli of mice by the mouse-passaged SARS-CoV-2 strain QHmusX.

### Infection of TMPRSS2-knockout mice with the Beta, Gamma and Omicron variants
Next, murine-susceptible SARS-CoV-2 variants were used to examine the role of TMPRSS2 after SARS-CoV-2 infection. The N501Y-carrying variants, including Beta and Gamma, show infection of WT mice[33]. Beta variant inoculation (TY8-612, $7.9 \times 10^4$ TCID$_{50}$ in 25 µl) resulted in more than 15% weight loss in WT mice but not in KO mice (Fig. 3a). Seroconversion was observed in all WT mice and half of the KO mice, and antibody titres were significantly lower in KO mice (Fig. 3a). The virus

titres in the lungs were significantly higher throughout the observation period in WT mice compared with KO mice (Fig. 3a). Similar histopathological differences were observed in the Beta variant-infected mice as in mice infected with mouse-passaged SARS-CoV-2 (Fig. 3b). Inoculation with the Gamma variant (TY7-503, $1.6 \times 10^6$ TCID$_{50}$ in 25 µl) resulted in less than 10% weight loss in WT mice, but no weight loss in KO mice (Fig. 3c). The seroconversion data, virus titres in the lungs and pathology were similar to those observed in experiments with mouse-passaged strains and the Beta variant (Fig. 3c, d).

Experimental infection of WT and KO mice was also performed with the Omicron variant (TY38-873, $1.0 \times 10^4$ TCID$_{50}$ in 25 µl), although the Omicron variant has been shown to have low virulence in mice[34]. The mice in both groups showed no body weight changes after intranasal inoculation and no seroconversion on day 9 p.i. (Fig. 3e). To evaluate the enzymatic activity of mouse TMPRSS2, Omicron S protein was expressed in HeLa cells constitutively expressing mouse TMPRSS2 or human TMPRSS2 (HeLa/mTMPRSS2 and HeLa/hTMPRSS2, respectively) together with human ACE2. The data showed that mouse TMPRSS2 and human TMPRSS2 were similarly active in inducing membrane fusion by the Omicron S protein (Supplementary Fig. 4). Indeed, the virus titres and viral RNA levels in the lungs and maxilla, including the nasal cavity, were higher in WT mice than in KO mice (Fig. 3f, g). In addition, viral subgenomic RNA levels revealed differences in viral replication in the airways of WT and KO mice (Fig. 3h). Finally, lung histopathology and immunohistochemical examination confirmed virus infection of the bronchi of WT mice in the early phase of infection (Fig. 3i). Inflammatory cells, mainly lymphocytes, infiltrated the viral antigen positive-peribronchial area in WT mice, but had cleared by day 9 p.i. Omicron-infected WT mice showed a few viral antigen-positive cells in the olfactory epithelium on day 3 p.i., without significant pathological changes, including inflammation (Fig. 3i). No viral antigen or lesions were observed in the airways of KO mice until day 9 p.i., with the exception of one KO mouse, and few antigen-positive cells were detected in the nasal cavity on day 5 p.i. These results indicated that N501Y-carrying SARS-CoV-2 variants, including Beta, Gamma and even Omicron infect murine airways in vivo primarily using TMPRSS2.

### Effect of nafamostat on SARS-CoV-2 infection in mice
The effect of nafamostat on virus growth in mice was analysed to validate further the data from knockout mice that TMPRSS2 plays an essential role in the growth of SARS-CoV-2 in mice. C57BL/6 mice were

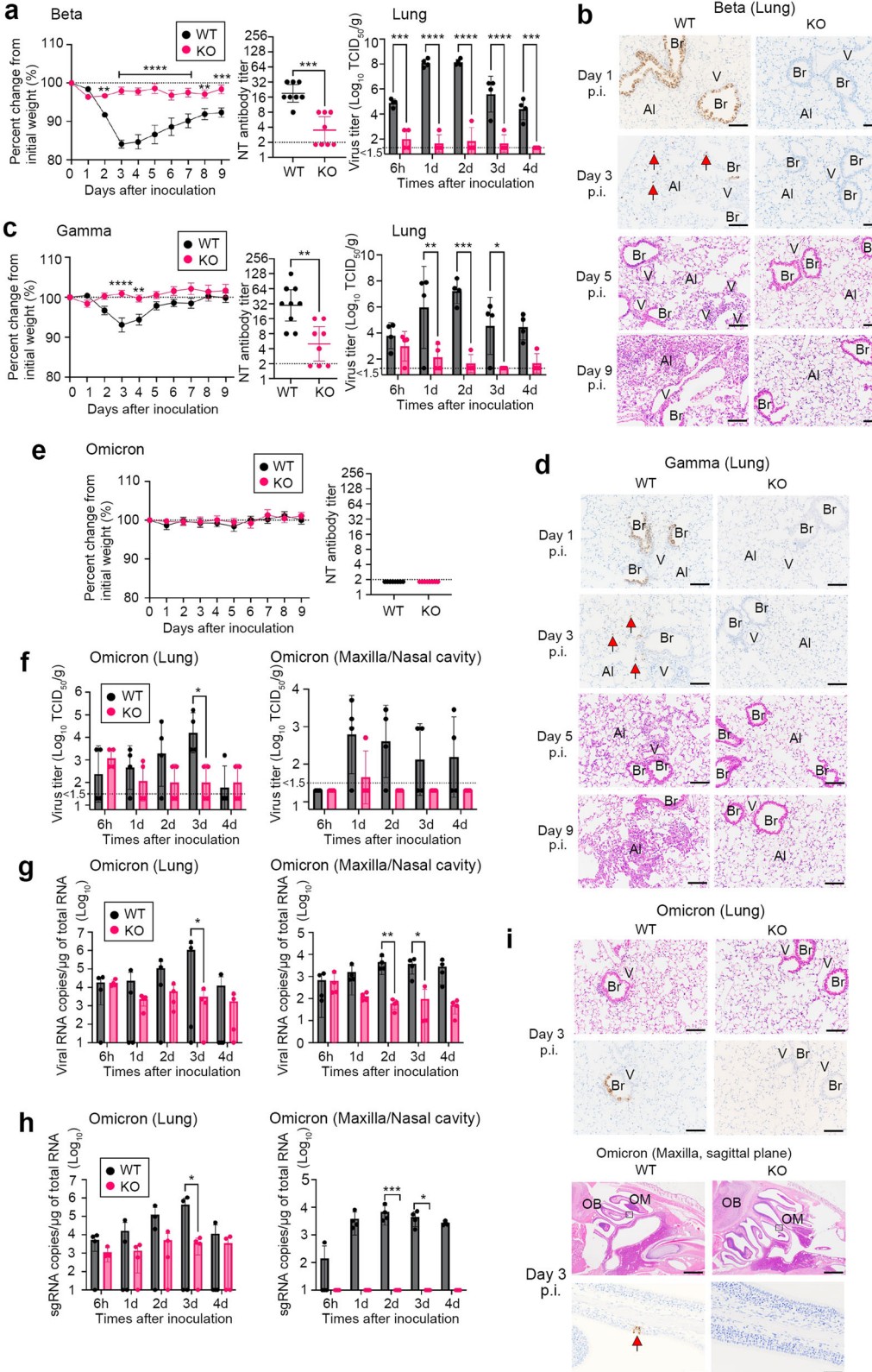

treated with intranasally nafamostat (100 μg/head) 2 h prior to infection and challenged with the Beta, Gamma and Omicron variants ($1 \times 10^4$ TCID$_{50}$/mouse) (Fig. 4a). In the nafamostat treatment group, infectious virus titres in the lungs were reduced 50- to 1000-fold for the Beta and Gamma variant (Fig. 4b). An approximately 10-fold inhibition of growth was observed in infection with the Omicron variant (Fig. 4b). However, this reduction was not statistically significant and

further experiments will be necessary to study effects of nafamostat on Omicron infection (Fig. 4b).

## Discussion

Many factors influence the transmissibility of SARS-CoV-2 and the disease severity resulting from infection. Among them, alterations in the function of the S protein of SARS-CoV-2 are significant in

**Fig. 3 | Experimental infection of TMPRSS2-knockout mice with SARS-CoV-2 variants.** C57BL/6 (WT) and TMPRSS2-knockout (KO) mice were inoculated intranasally with Beta (TY8-612) (**a**, **b**), Gamma (TY-503) (**c**, **d**) and Omicron (TY38-873) variants (**e–i**). (**a**, **c**, **e** left panels) Body weight curve during the observation period a: $n = 8$ (4 males and 4 females for KO and WT); **c** $n = 8$ (KO, 4 males and 4 females), $n = 9$ (WT, 3 males and 5 females); **e** $n = 8$ (4 males and 4 females for KO and WT); **a** **$*P = 0.0037$ at 2 days post-infection (p.i.) and 0.0019 at 8 days p.i., ***$P = 0.0002$ at 9 days p.i. and ****$P < 0.0001$ from 3 to 7 days p.i.; **c** ****$P < 0.0001$ at 3 days p.i. and **$*P = 0.0060$ at 4 days p.i., by two-way ANOVA followed by Bonferroni's multiple comparison test. Data are mean ± SEM. (**a**, **c** middle panels and **e** right panel) Serum neutralisation titres (NT) at 9 days p.i., respectively. The dashed line indicates the limit of detection (<2) **a** $n = 8$ (4 males and 4 females for KO and WT); **c** $n = 8$ (KO, 4 males and 4 females), $n = 9$ (WT, 3 males and 5 females); **e** $n = 8$ (4 males and 4 females for KO and WT); **a** ***$P = 0.0006$; **c** **$*P = 0.0028$ by two-tailed Mann–Whitney U test. Data with a geometric mean (GMT) ± 95% confidence interval, CI (**a**, **c** right panels and **f**). Viral titres in lung homogenates or nasal cavity at 6 h and 1 to 4 days p.i. ($n = 4$ per group, 2 males

and 2 females). The detection limit was $10^{1.5}$ $TCID_{50}/g$ of tissue. **a** ***$P = 0.0004$ at 6 h and $P = 0.0002$ at 4 days p.i. and ****$P < 0.0001$; **c** *$P = 0.0226$ at 3 days p.i.), **$*P = 0.0064$ at 1 day p.i. and ***$P = 0.0002$ at 2 days p.i.; **f** *$P = 0.0147$ at 3 days p.i. by two-way ANOVA followed by Bonferroni's multiple comparison test. Data are mean ± SD. (**b**, **d**, **i**) Histopathology of the respiratory tract from WT and TMPRSS2-KO mice infected with Beta, Gamma and Omicron variants. Immuno-histochemical analysis (IHC) using an anti-SARS-CoV-2 NP antibody at 1 and/or 3 days p.i ($n = 4$ per group, 2 males and 2 females). Haematoxylin and eosin staining (HE) at 3, 5, or 9 days p.i. Br, bronchi; V, vein; Al; alveoli; OB, olfactory bulb; OM, olfactory mucosa. Bars, 50 or 100 μm. Red arrows indicate virus antigens (**g**, **h**) The levels of viral RNA (**g**) and subgenomic viral RNA (**h**) in the lung homogenates or nasal cavity at different time points after inoculation ($n = 4$ per group, 2 males and 2 females). Each gene was normalised to that of total RNA. Samples were from the same experiment as those in (**g**). **g** left panel: *$P = 0.0184$; **g** right panel: *$P = 0.0272$ and **$*P = 0.0055$; **h** left panel: *$P = 0.0165$; **h** right panel: *$P = 0.0221$ and ***$P = 0.0006$ by two-way ANOVA followed by Bonferroni's multiple comparison test. Data are mean ± SD.

determining the host range, infectivity, target cells and tissues to be infected and pathogenicity of the virus. S1/S2 cleavage by furin promotes S2′ cleavage by TMPRSS2, which triggers the virus to enter a cell through the plasma membrane[3,4]. It was speculated that the efficient use of this early entry pathway contributes to improved transmissibility in humans[4]. If correct, this could partly explain the increase in transmissibility of the D614G mutant and the Alpha to Delta variants. However, several studies[24–27] demonstrated reduced cleavage of the Omicron S protein and along with other studies[24,27–30] concluded that the Omicron variant poorly utilises TMPRSS2 and mainly enters cells through the endocytosis pathway. However, there are caveats to the interpretation of these data. Much of the data on the Omicron variant from the previous studies[24,27,28,35] is presented as a comparison with the Delta variant, which has a highly cleavable S protein and superior fusion capacity among the SARS-CoV-2 variants. In addition, data using pseudotype viruses as a proxy for authentic SARS-CoV-2 have often been the basis from which conclusions have been drawn and indeed, TMPRSS2-independence or inefficient S protein cleavage of Omicron is less evident when using authentic SARS-CoV-2 strains than pseudotype viruses[24,30].

Our data showed that the Omicron S protein expressed from authentic SARS-CoV-2 is cleaved to at least the same extent or even more than the S proteins of the Alpha, Beta and Gamma variants. Lamers et al.[30] also showed a comparable level of S protein cleavage in the Omicron variant compared with other variants. In our data, unimpaired cleavage of the Omicron S protein was evident when compared with the Wuhan-type S protein. In the data reported by Shuai et al.[27], the decrease in the cleavage rate of the Omicron variant was also less pronounced. Therefore, we propose that interpretation of the transmissibility of the Omicron variant based on inefficient cleavage of the S protein may need some modification. Lamers et al.[30] suggested that the Omicron variant may utilise serine proteases other than TMPRSS2 since Omicron infection is not enhanced by TMPRSS2 but is inhibited by camostat, but not EST. In any case, all studies, including our present study, agree with each other in the following respect[24,26,28–30,35]. The TMPRSS2-utilisation ability and cell fusion ability of the Omicron S protein is reduced and this may play a role in the attenuated phenotype of the Omicron variant.

Structural analyses of the Omicron S protein have provided an essential clue to understanding the reduced fusion ability and TMPRSS2-usage by Omicron. The Omicron S protein has a compact structure with reduced distances between domains and protomers and it is not easy for the RBD to move into the up position required for receptor binding[36,37]. Receptor binding is also a prerequisite for efficient use of TMPRSS2[5]. Therefore, it would be reasonable that a more significant number (~10 times higher number) of receptor molecules (ACE2) is required for the Omicron S protein to exert membrane fusion activity[38].

In most cases, TMPRSS2 enhances SARS-CoV-2 infection in vitro, but these results do not prove that TMPRSS2 is used in actual infection or that it is involved in pathogenesis. TMPRSS2 can also facilitate the early entry pathway of SARS-CoV from the cell surface. However, SARS-CoV does not have a furin cleavage motif at the S1/S2 site and its TMPRSS2-dependency in cultured cells is not as clear as that of SARS-CoV-2. Indeed, infection of cultured cells by SARS-CoV can be achieved very efficiently only by the cathepsin-mediated endocytosis pathway. Nevertheless, in a mouse model[39], infection of SARS-CoV cannot be inhibited by cysteine protease inhibitors that block the cathepsin-mediated endocytosis pathway, while the infection is attenuated by camostat, which inhibits serine proteases such as TMPRSS2. In agreement with these data[39], we have previously demonstrated that TMPRSS2 plays an essential role in virus spread within the airways of a murine model infected with SARS-CoV and Middle East respiratory syndrome (MERS)-CoV[40]. The reduction in viral load due to the lack of TMPRSS2 expression was moderate for these viruses, ranging from 1/10 to 1/100. However, pathological changes were evident. The data revealed that TMPRSS2 is significantly involved in SARS-CoV and MERS-CoV infection and inflammation in the airways. In this study, experimental infection with a mouse-passaged strain and N501Y-carrying variants of SARS-CoV-2, including Beta, Gamma and Omicron, revealed that deficiency of TMPRSS2 caused minimal viral growth and induced no pathological lesions in the lungs. These data indicated that TMPRSS2 has a more important role during SARS-CoV-2 infection than during infection of SARS-CoV and MERS-CoV in mouse models.

Like SARS-CoV, in vitro experiments have reported that the Omicron variant enters cells in a cathepsin-dependent manner[24]. However, our in vivo experiments using TMPRSS2-KO mice and a serine inhibitor in this study suggest that the Omicron variant, like SARS-CoV[39] and other SARS-CoV-2 strains, utilizes a furin/TMPRSS2-dependent entry pathway rather than the endocytosis pathway for the infection of the airways, at least in murine models. These results will have significant implications for the elucidation of the pathogenesis of SARS-CoV-2 and the development of therapeutic strategies.

The Omicron variant failed to propagate in human lung epithelial Calu-3 cells, in which the TMPRSS2-mediated early entry pathway is used. If TMPRSS2-usage is critical for SARS-CoV-2 infection in vivo, this may explain the attenuated phenotype of the Omicron variant[26,27,34,41], but cannot explain the high transmissibility of the Omicron variant. A key message from this study is that the Omicron variant is not an exception in using TMPRSS2 in murine models and TMPRSS2 is a critical virulence factor even for the Omicron variant. Of course, there may be discrepancies in the results obtained with mice and the pathology in humans; however, the findings from our in vivo study contribute toward understanding the pathogenesis of SARS-CoV-2, including the Omicron variant.

## a

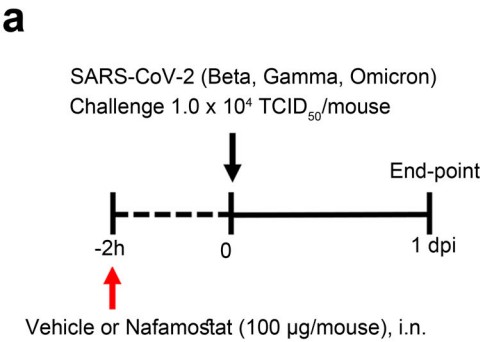

SARS-CoV-2 (Beta, Gamma, Omicron)
Challenge 1.0 × 10⁴ TCID₅₀/mouse

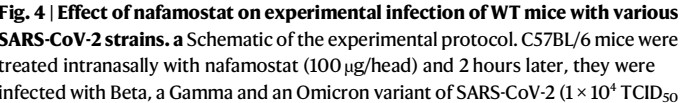

Vehicle or Nafamostat (100 μg/mouse), i.n.

## b

Virus titer (1 dpi)

**Fig. 4 | Effect of nafamostat on experimental infection of WT mice with various SARS-CoV-2 strains. a** Schematic of the experimental protocol. C57BL/6 mice were treated intranasally with nafamostat (100 μg/head) and 2 hours later, they were infected with Beta, a Gamma and an Omicron variant of SARS-CoV-2 (1 × 10⁴ TCID₅₀

in 25 μl) **b** Lung virus titres measured at 1 day post-infection ($n = 6$ mice per group). The detection limit was $10^{1.5}$ TCID₅₀/g of tissue. $*P = 0.0195$, $**P = 0.0065$, $P = 0.2273$ (Omicron) by two-tailed Mann–Whitney U test. Data are mean ± SD.

# Methods

## Ethics statement

All procedures involving cells and animals were conducted in a bio-safety level (BSL) three laboratory. Experiments using recombinant DNA and pathogens were approved by the Committee for Experiments using Recombinant DNA and Pathogens at the National Institute of Infectious Diseases, Tokyo, Japan (approval no. 2-84). All animal experiments were approved by the Animal Care and Use Committee of the National Institute of Infectious Diseases in Japan (approval no. 120153-II, 122045-II). All experimental animals were handled in BSL3 animal facilities according to the guidelines of this committee (approval no. 21-84, 22-53). All animals were housed in a facility certified by the Japan Health Sciences Foundation. The isolation of virus from clinical specimens was approved by the Medical Research Ethics Committee of the National Institute of Infectious Diseases (NIID) for the use of human subjects (approved ID: 1091).

## Cells

VeroE6/TMPRSS2 cells were generated in our laboratory previously[9,42] and deposited in the Japanese Collection of Research Bioresources Cell Bank (the National Institute of Biomedical Innovation, Health and Nutrition, Osaka, Japan). Cells were cultured in Dulbecco's modified Eagle's medium (DMEM), low glucose (Sigma-Aldrich, St. Louis, MO, USA), containing 10% foetal calf serum (FCS) and antibiotics (100 U/ml penicillin and 0.1 mg/ml streptomycin). The cells were maintained in the presence of 1.0 mg/ml geneticin (G418; Nacalai Tesque) and were used for analysis without geneticin. Vero cells (ATCC CCL-81) were cultured in DMEM with 5% FCS and antibiotics. The human bronchial epithelial Calu-3 cells were purchased from ATCC and cultured in DMEM with 10% FCS and antibiotics. This study used a subculture lot (NIIDv3 lot) of Calu-3 cells, which have been passaged for dozens of generations in our laboratory and have improved proliferation ability. HeLa cells stably expressing human TMPRSS2 (HeLa/hTMPRSS2) and mouse TMPRSS2 (HeLa/mTMPRSS2) were generated in our laboratory previously[31]. Canine kidney cell line (MDCK cell line) was originally obtained from Dr. Peter Palese in Icahn School of Medicine at Mount Sinai.

## Viruses

Viruses were isolated from anonymized clinical specimens (naso-pharyngeal/nasal swabs or saliva) collected from individuals diagnosed with COVID-19 as part of the public health diagnostic activities conducted by NIID[9,43]. VeroE6/TMPRSS2 cells were inoculated with the specimens, cultured in DMEM supplemented with 2% FCS and anti-biotics and observed daily for the appearance of cytopathic effects (CPEs)[9,43]. If necessary, the cells were passaged several times until

apparent CPEs were observed. The culture supernatants containing isolated viruses were stored at −80 °C. The nearly full-length genome sequences of isolated viruses were determined as reported previously[43]. The data were deposited in the Global Initiative on Sharing All Influenza Data (GISAID) database. The clinical isolates used in this study are shown in Supplementary Table 1. The mouse-passaged SARS-CoV-2 QHmusX strain was reported previously[44]. A/Vietnam/1194/04 (VN1194 [H5N1]) and A/Anhui/1/2013 (Anhui1 [H7N9]) influenza viruses were reported previously[31].

## Virus titration

The infectivity titres of SARS-CoV-2 were analysed using VeroE6/TMPRSS2 cells cultured in DMEM containing 2% FCS and antibiotics. Viral infectivity titres were expressed as the 50% tissue culture infectious dose (TCID₅₀) per millilitre calculated according to the Behrens–Kärber method. All experiments with infectious SARS-CoV-2 were performed under BSL3 conditions. The infectivity titres (plaque-forming units [PFUs]) of influenza viruses were analysed[31].

## Plaque assay on VeroE6/TMPRSS2 and Vero cells

One day before infection, VeroE6/TMPRSS2 and Vero cells (1.0 × 10⁶ cells) were seeded into six-well plates and infected with virus samples. After 1 h of incubation at 37 °C, the viruses were removed, and the cells were washed and overlaid with DMEM containing 2% foetal bovine serum (FBS) and 0.8% SeaPlaque™ agarose (Lonza 50100). At 3 days after infection, the cells were fixed with 3.7% buffered for-maldehyde and stained with 1% crystal violet. To assess the effect of protease inhibitors [(2 S,3 S)-trans-Epoxysuccinyl-ʟ-leucylamindo-3-methylbutane ethyl ester (EST) (330005: Calbiochem, San Diego, CA, USA)] on plaque formation, we pretreated the cells with EST for 30 min. After pre-treatment, the cells were washed twice and sub-jected to a plaque assay using DMEM containing 2% FBS and 0.8% SeaPlaque™ agarose, and EST. Then, cell staining was performed, as described above.

## Protease inhibition assay on Calu-3 cells

One day before infection, Calu-3 cells (1 × 10⁵ cells/well) were seeded into a 96-well plate. To assess the effect of protease inhibitors [EST or nafamostat (Tokyo Chemical Industry, Tokyo, Japan)] on SARS-CoV-2 infection of Calu-3 cells, we pretreated the cell monolayers with these inhibitors for 30 min at 37 °C. Cells were then inoculated with SARS-CoV-2 (MOI 0.1 or 1) and incubated with the inhibitors for 1 h at 37 °C. The residual virus was removed, and fresh medium containing inhibi-tors at the indicated concentrations was added to the cells, followed by culturing at 37 °C for 23 h. Cellular RNA was isolated using the QIAamp 96 Virus QIAcube HT kit and QIAcube (Qiagen, Hilden, Germany).

Then, a real-time PCR assay was performed to ascertain the amount of newly synthesised subgenomic SARS-CoV-2 RNA.

## Real-time RT-qPCR

Real-time RT-qPCR was performed to detect SARS-CoV-2 RNA using the QuantiTect Probe RT–PCR kit (Qiagen) with the following primers and probes: NIID_2019-nCOV_N_F2 (5′-AAATTTTGGGGACCAGGAAC-3′), NIID_2019-nCOV_N_R2 (5′-TGGCAGCTGTGTAGGTCAAC-3′), and NIID_2019-nCOV_N_P2 (5′-FAM-ATGTCGCGCATTGGCATGGA-BHQ′-3′), targeting the N gene, SARS-CoV-2_NIID_S_F1 (5′-CAGTCAGCACCTCATG GTGTA-3′), SARS-CoV-2_NIID_S_R3 (5′-AACCAGTGTGTGCCATTTGA-3′), and SARS-CoV-2_NIID_S_P2 (5′-FAM-TGCTCCTGCCATTTGTCATGATG G-BHQ1-3′), SARS2-LeaderF60 (5′-CGATCTCTTGTAGATCTGTT CTCT-3′), SARS2-N28354R (5′-TCTGAGGGTCCACCAAACGT-3′), and SARS2-N28313Fam (5′-FAM-TCAGCGAAATGCACCCCGCA-TAMRA-3′) for targeting the subgenomic RNA. First, the reaction mixtures were incubated at 50 °C for 30 min, followed by incubation at 95 °C for 15 min, and thermal cycling, which consisted of 40 cycles of denaturation at 94 °C for 15 s, and annealing and extension at 60 °C for 60 s. The assay was performed on a LightCycler 480 (Roche, Basel, Switzerland). Data analysis was performed using LightCycler 480 software (version 1.5.1) (Roche). The detection rate (equivalence) of these primer-probe sets for variants, including Omicron, has already been reported[45].

## SARS-CoV-2 infection (cultured cell experiment)

One day prior to infection, VeroE6/TMPRSS2 cells ($1 \times 10^4$ cells) and Calu-3 cells ($1 \times 10^4$ cells) were seeded into a 96-well plate. First, SARS-CoV-2 was inoculated at an MOI of 0.1 and incubated at 37 °C for 1 h. Then, the infected cells were washed, and 100 μl of culture medium was added. The culture supernatant (5 μl) was harvested at the indicated time points, mixed with 45 μl nuclease-free water, and incubated at 99 °C for 5 min. The amount of viral RNA was then quantified by real-time PCR.

## SDS-PAGE and immunoblotting

VeroE6/TMPRSS2 cells were infected with SARS-CoV-2 (MOI 0.01 or 0.1). After 24 and 48 h, the culture medium was harvested and centrifuged, and the supernatants were collected. The supernatants were diluted with 4 × sample buffer (Bio-Rad, Hercules, CA, USA) and boiled for 10 min. For cell lysates, the harvested cells were washed and lysed in RIPA buffer (FUJIFILM Wako Pure Chemical Corporation, Osaka, Japan) containing a complete protease inhibitor mixture (Roche Diagnostics), and lysates were diluted with 4 × sample buffer and boiled for 10 min. The polypeptides were separated by SDS-PAGE and immunoblotted. For protein detection, the following antibodies were used: mouse anti-SARS-CoV-2 S1 subunit monoclonal antibody (clone #1035206, R&D systems, Minneapolis, MN, USA, Cat# MAB105403, 1:500), rabbit anti-SARS-CoV-2 S2 subunit polyclonal antibody (Abcam, Cambridge, UK, Cat# ab272504, 1:1000), an in-house rabbit anti-SARS-CoV-2 N antibody was used as the primary antibody[44], horseradish peroxidase (HRP)-conjugated F(ab′)₂ fragment of affinity purified anti-mouse IgG [H&L] [Goat] antibody (ROCKLAND, Cat#710-1332, 1:5000) and HRP-conjugated goat anti-rabbit IgG polyclonal antibody (MP Biomedicals, Irvine, CA, USA, Cat# 55689, 1:5000). Chemiluminescence was detected using SuperSignal West Femto Maximum Sensitive HRP Substrate (Thermo Fisher Scientific, Waltham, MA, USA). Polypeptides (the uncleaved S, S1, S2 and other S protein-specific small size products) were detected using Amersham Imager 800 (Cytiva, Tokyo, Japan), and the band intensity was quantified with Fiji software v2.2.0 (ImageJ).

## Plasmid construction and syncytium formation assay

The Wuhan-type S protein expression plasmid (VG40589-UT) was purchased from SinoBiological (Beijing, China) and the S protein open reading frame was cloned into pCAGGS vector. The cDNAs of the Alpha, Beta, Gamma, Delta, and Omicron S genes were reverse-transcribed from purified viral RNAs of strains hCoV-19/Japan/ QHN001/2020, hCoV-19/Japan/TY8-612/2021, hCoV-19/Japan/TY7-501/ 2021, hCoV-19/Japan/TY11-927-P1/2021 and hCoV-19/Japan/TY38-873P0/2021, respectively, and cloned into pCAGGS vector. VeroE6/ TMPRSS2 cells were transfected with these S protein-expressing plasmids using Lipofectamine LTX (Invitrogen, Waltham, MA, USA). Human ACE2 cDNA was amplified from the human lung cDNA library (CLONTECH, Palo Alto, CA, USA) using oligonucleotides specific for human ACE2 (ACE2f: 5′-ATC TTG GCT CAC AGG GGA CGA TG-3′ and ACE2r: 5′-ACC TCA AGA GGA AAA ACA TAG A-3′). The resulting PCR product was inserted into pTargeT vector (Promega, Madison, WI, USA). HeLa/mTMPRSS2, HeLa/hTMPRSS2, and the parental HeLa cells were transfected with 0.5 μg of Omicron S protein-expressing plasmid (MC_0101274, GenScript Biotech, Piscataway, NJ, USA) together with 0.5 μg of the human ACE2-expressing plasmid using X-tremeGENE HP transfection reagent (Roche). At 2 days post-transfection, the cells were washed with phosphate-buffered saline (PBS) and fixed with 10% buffered formalin solution. For Syncytia detection, the following antibodies were used: rabbit anti-SARS-CoV-2 S polyclonal antibody (Proteintech, Rosemont, IL, USA, Cat#28867-1-AP, 1:1000) and Alexa 549-conjugated Goat anti-rabbit IgG (H + L) Cross-Adsorbed secondary antibody (Thermo Fisher Scientific, Cat#A-11012, 1:2000). The nuclear DNA was stained with 4′,6′-diamidino-2-phenylindole (DAPI). The images were acquired using All-in-One Fluorescence Microscope BZ-X800 (Keyence, Osaka, Japan). The fusion index was calculated as the average number of nuclei per syncytium.

## Mice

C57BL/6 mice were purchased from Japan SLC (Shizuoka, Japan) and were maintained in specific-pathogen-free facilities. TMPRSS2-KO mice established from TMPRSS2 gene knockout C57BL/6 embryonic stem cells (product number VG13341), which were obtained from the Knockout Mouse Project (KOMP) Repository (UC Davis), were propagated and maintained for the animal study[39]. Mice were divided into groups of five to six and housed in rooms with controlled and monitored ambient temperature (24 °C) and humidity (40–60%) with 12-hour day/night lighting.

## Animal study design

C57BL/6 mice and TMPRSS2-KO mice were intranasally inoculated with 25 μl of SARS-CoV-2. The inoculating viral titres per animal were as follows: the mouse-passaged SARS-CoV-2 of strain QHmusX, $1.9 \times 10^4$ TCID$_{50}$; the Beta strain of TY8-612, $7.9 \times 10^4$ TCID$_{50}$; the Gamma strain of TY7-503, $1.6 \times 10^6$ TCID$_{50}$; and the Omicron strain of TY38-873, $1.0 \times 10^4$ TCID$_{50}$. Body weight was measured daily for 9 days ($n = 8$–10 per group), and animals were euthanized at 6 h and at 1, 2, 3, 4, 5 and 10 days p.i. to analyse viral replication, cytokine expression, and/or disease pathology ($n = 4$ or 8–10 per group). Clinical signs were observed up until 9 days p.i. The humane endpoint was defined as the appearance of clinical diagnostic signs of respiratory stress, including respiratory distress and more than 25% weight loss. Animals were euthanized under anaesthesia with an overdose of isoflurane if severe disease symptoms or weight loss were observed. In these experiments, male and female mice of both strains were used between 14- and 34-week-old. Six C57BL/6 mice per group were treated intranasally with nafamostat (100 μg/25 μl/head) or vehicle alone (PBS). Two hours later, mice were infected intranasally with $1 \times 10^4$ TCID$_{50}$ in 25 μl of a Beta, a Gamma and an Omicron variant of SARS-CoV-2. Twenty-four-week-old female mice were used in this experiment. Virus titres were measured in harvested lungs by TCID$_{50}$ 1 day after infection. C57BL/6 mice and TMPRSS2-KO mice were intranasally inoculated with $1.0 \times 10^4$ PFU (20 μl) of H7N9 (Anhui1) or H5N1 (VN1194) of influenza virus. In this experiment, male and female mice of both strains were used 9-week-

old. Body weight was measured daily for 4 days ($n = 6$ per group) and animals were euthanized to analyse viral replication ($n = 6$ per group). The humane endpoint was defined as the appearance of clinical diagnostic signs of respiratory stress, including respiratory distress and more than 25% weight loss. To determine the infectious virus titres in vivo, the lung was washed twice by injecting a total of 2 ml PBS containing 0.1% bovine serum albumin, and bronchoalveolar lavage fluid was collected to analyse infectious virus titres. The virus titre was determined with plaque assay[31]. The virus titre was calculated as plaque-forming units (pfu) per ml (pfu/ml).

### RNA extraction and quantification of viral RNAs in the lung homogenate
Viral RNA from each lung homogenate was isolated using TRIzol reagent (Thermo Fisher Scientific) and a benchtop automated DNA/RNA extraction machine (Maxwell RSC48, Promega, Madison, WI, USA) following the manufacturer's suggested protocol. RNA was quantified by a NanoDrop spectrophotometer (Thermo Fisher Scientific). Viral RNA quantification of each sample was performed by real-time RT-qPCR.

### SARS-CoV-2 neutralising assay
Blood was obtained from each mouse and allowed to clot. Sera were then obtained by centrifugation and were inactivated by incubation at 56 °C for 30 min. Aliquots (100 $TCID_{50}$) of SARS-CoV-2 were incubated for 1 h in the presence or absence of mouse serum (serially diluted two-fold) and were then added to confluent VeroE6/TMPRSS2 cell cultures in 96-well microtiter plates. Samples were examined for viral CPEs on day 5 and the neutralising antibody titres were determined as the reciprocal of the highest dilution at which no CPE was observed. The lowest and highest serum dilutions tested were 1:4 and 1:512, respectively. The average geometric mean titre (GMT) was calculated using GraphPad Prism 9 software (GraphPad Software, La Jolla, CA, USA).

### Histopathology and immunohistochemistry
To obtain animal tissues, we anesthetized and perfused mice with 2 ml of 10% phosphate-buffered formalin, and the lungs and head, including the nasal cavity and brain, were harvested and fixed. Fixed tissues were routinely embedded in paraffin, sectioned and stained with haematoxylin and eosin (H&E). For immunohistochemistry, antigen retrieval of formalin-fixed mouse tissue sections was performed by autoclaving at 121 °C for 10 min in retrieval solution at pH 6.0 (Nichirei, Tokyo, Japan). SARS-CoV-2 antigens were detected using a polymer-based detection system (Nichirei-Histofine Simple stain mouse MAX PO; Nichirei Biosciences, Inc., Tokyo, Japan), and an in-house rabbit anti-SARS-CoV-2 N antibody was used as the primary antibody[44]. Diaminobenzidine (Sigma-Aldrich) was used as the chromogen for HRP visualisation. Nuclei were counterstained with haematoxylin for 10 s. The images were acquired using the imaging software cellSens Standard 2.1 (Olympus).

### Detection of inflammatory cytokines and chemokines in mice
Homogenised mouse lung tissue samples (10% w/v) were diluted 1:1 in cell extraction buffer [10 mM Tris, pH 7.4, 100 mM NaCl, 1 mM EDTA, 1 mM EGTA, 1 mM NaF, 20 mM $Na_4P_2O_7$, 2 mM $Na_3VO_4$, 1% Triton X-100, 10% glycerol, 0.1% SDS and 0.5% deoxycholate (BioSource International, Camarillo, CA, USA)], incubated for 10 min on ice with vortexing, irradiated for 10 min with UV-C light to inactivate infectious virus, and tested in the BSL2 laboratory. Cytokine and chemokine levels were measured with a commercial mouse cytokine–chemokine magnetic bead panel 96-well plate assay kit (Milliplex MAP kit, Merck Millipore, Burlington, MA, USA), which detects 21 cytokines and chemokines: eotaxin, IFN-γ, IL-1α, IL-1β, IL-2, IL-4, IL-5, IL-6, IL-10, IL-12 (p40), IL-12 (p70), IL-13, IL-17, IP-10/

CXCL10, KC/CXCL1, MCP-1/CCL2, MIP-1α/CCL3, GMCSF, MIG/CXCL9, RANTES/CCL5 and TNF-α. The assay samples were read on a Luminex 200 instrument with xPONENT software (version 4.2, Merck Millipore), as described by the manufacturer.

### Statistical analyses
All data are expressed as the mean and standard error of the mean or mean and standard deviation, except for neutralising antibodies (GMT + 95% confidence interval, CI). Statistical analyses were performed using GraphPad Prism 9 software (version 9.4.1). Intergroup comparisons were performed using nonparametric analysis. A $P$ value <0.05 was considered statistically significant. The relevant figure legends indicate the number of independent experiments and technical replicates used. Statistical analyses included unpaired $t$-tests, Mann–Whitney U tests and ANOVA with multiple correction post-tests.

### Reporting summary
Further information on research design is available in the Nature Research Reporting Summary linked to this article.

## Data availability
All data supporting the findings of this study are available within the paper and in the Source Data. There are no restrictions in obtaining access to primary data. The sequences of the virus isolates used are available in the Global Initiative on Sharing All Influenza Data (GISAID) database. The list of virus names used in this study and GISAID ID numbers are shown in Supplementary Table 1. The cell line information of VeroE6/TMPRSS2 is available from JCRB Cell Bank in Japan (https://cellbank.nibiohn.go.jp/english/) (JCRB no. JCRB1819). Source data are provided with this paper.

## Code availability
No code was used in the course of the data acquisition or analysis.

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

## Acknowledgements

We are grateful to Ms. Midori Ozaki, Ms. Takiko Yoshida, Dr. Dai Izawa, and Ms. Miyuki Ban for their technical assistance and Dr. Akira Ainai and, Dr. Hideki Hasegawa, and our colleagues at the Institute for helpful discussions. We also thank the members of the Management Department of Biosafety and Laboratory Animals for support with the BSL3 facility. We thank Dr. Yoshihiro Kawaoka (The University of Tokyo) for providing a SARS-CoV-2 Omicron variant; Dr. Yuelong Shu, Chinese Center for Disease Control and Prevention, and Dr. Le Quynh Mai, National Institute of Hygiene and Epidemiology, for providing H7N9 and H5N1 Influenza A virus, respectively. This work was supported by the Japan Agency for Medical Research and Development (AMED) under grant numbers JP21fk0108615 and JP 20fk0108411 to N.N., by the Ministry of Health, Labour and Welfare under grant numbers 20HA2007 and 21HA2003 to M.T., and the Grant-in-Aid for Scientific Research from the Ministry of Education, Culture, Sports, Science, and Technology in Japan under grant numbers 18H02665 to M.T. and N.N., 19K08945 to N.I-Y., and 20K21666 to N.N. This work was also supported by grants from the Research Foundation for Opto-Science and Technology, The Naito Foundation, and Japanese Respiratory Foundation to M.T. We thank Edanz (https://jp.edanz.com/ac) for editing a draft of this manuscript.

## Author contributions

Conceived research and designed study: N.I.-Y., M. Kakizaki, K.S., N.N., M. Takeda; designed experiments: N.I.-Y., M. Kakizaki, K.S., N.N., M. Takeda; performed animal experiments: N.I.-Y., N.N., H.A.; performed in vitro experiments: M. Kakizaki, T.O., M. Tahara, M. Kawase, H.A., Y.T., I.T.; prepared and provided SARS-CoV-2 isolates: S.F., K.M.; prepared and provided Influenza viruses: H.A.; analysed data: N.I.-Y., M. Kakizaki, N.S.-S., S.M., K.S., T.S., N.N., M. Takeda; funding acquisition: N.I.-Y., T.S., N.N., M. Takeda, and drafted the manuscript: N.N., M. Takeda. All authors contributed to the editing of the manuscript.

## Competing interests

The authors declare no competing interests.
