## [Peer Review File · Nature Communications]

Essential role of TMPRSS2 in SARS-CoV-2 infection in murine airwaysREVIEWER COMMENTS

Reviewer #1 (Remarks to the Author):

This submission makes an important observation that helps to clarify some findings in the SARS-CoV-2 / COVID19 field. It shows that the recent SARS-CoV-2 VOCs, particularly omicron, has TMPRSS2 as a host susceptibility factor in laboratory mice. This is important because in vitro (cell culture) results demonstrate that the omicron VOC bypasses the TMPRSS2 activating proteases and instead uses the "late" endosomal proteases for spike-mediated fusion activation. The cell culture results are clearly not able to provide a full picture of the SARS-CoV-2 variants and their utilization of cell surface serine TMPRSS proteases in many infection contexts.

The authors of this submission repeated many of the previously published cell culture findings, and they advanced the cell culture results modestly by including more VOCs in their analysis. This work is solid and convincing. The new findings are with the TMPRSS2 KO mice and this is what gives significance to the report. These findings are valuable for the field, especially because TMPRSS inhibitors such as camostat and nafamostat are promoted as clinically useful antivirals to treat COVID19. However, at present there are some possible caveats to the exclusive use of TMPRSS2 KO mice for analysis. Also, there are some aspects of the manuscript that need to be reconsidered. There are major and minor comments below:

Major comments:

1. It would help if the investigators could challenge WT and KO mice with a respiratory virus that is known to be completely TMPRSS2-independent. This would provide a control to determine whether the TMPRSS2 KO condition is broadly incapable of supporting a respiratory virus infection (i.e., are there pleiotropic effects of the TMPRSS2 KO condition that render the lung environment less susceptible to many viruses).
2. If a TMPRSS2-independent virus is not known or available, then it would help if the investigators evaluated whether nafamostat or camostat suppress the VOC infections in WT mice, down to levels equivalent to those observed in the TMPRSS2 KO mice. This orthogonal pharmacologic approach may be necessary to fully validate the genetic data derived from the TMPRSS2 gene KO condition.
3. Discussion section appears to have biases and some statements do not appear to be adequately supported. For example, lines 249-250, and lines 254-255: IT is really not clear that the VLPCs are "driven by the TMPRSS2 pathway" and it is not clear whether adaptations to the early entry pathway "explain the increase in transmissibility of D614G, alpha and delta. There are many other selective forces at play in determining transmissibility. The discussion section requires several revisions.
4. A key reference is missing; PMID: 25666761. This paper shows that SARS-CoV, a virus that definitely uses the late cathepsin pathway in cell culture, does in fact use the camostat-sensitive TMPRSS pathway in laboratory mice. It should be cited and discussed.

Minor comments:

1. Lines 169-170; "the cathepsin pathway cannot be a substitute for the TMPRSS2 pathway": This seems to be a significant important conclusion but is not explained or discussed in any way. What are the authors' speculations about this?
2. Fig 1A; cannot see the SD ranges, and furthermore the P values as noted in the Fig 1 legend are not assigned by asterisks.
3. Fig 1B; are we looking at duplicate results; why no gamma in the second lower set of results?
4. Fig 1D; EST or E64d? Are they the same? If the same, then suggest consistence between figure and figure legend.

5. Fig 2B; define "crossing point"; is this the same as "cycle threshold"? (I am not familiar with "crossing point").
6. Figs 3 and 4 are essentially one figure.

Reviewer #2 (Remarks to the Author):

This is a timely article that addresses an important topic – the activation of SARS-CoV-2 Omicron spike and the entry of the virus into cells. The authors portray a slightly different story to the prevailing view that Omicron has a very different entry pathway to other variants. The current work suggests a shift in pathway, but no fundamental change in the activating protease (TMPRSS2).

A general comment is that SARS-CoV-2 entry is complex and variable and it is easy to draw inappropriate conclusions. With this in mind I would suggest the authors be more careful with wording in the title and abstract – eg saying that multiple entry pathways exist, rather than there two...." – recent studies have indicated (rather than 'shown'). What is meant by "improved" use (line 35), also "not impaired" (line 36). They should specify cell types in the abstract (line 37). It is also worth considering the use of "in vivo" in the Title – as this really refers to mice

The manuscript would be improved by better annotation of the figures (headings for cell types etc).

Figure 1B is very unclear – it is not labelled well, and there seems to be a missing lane for the lower panel (was gamma not tested?). The bar charts in panel C are very hard to visualize, and statistics need to be clearer (what is being compared with the **). It seems that Omicron spike is less well incorporated into particles (is this true?) but cleaved once it is present?

Figure 3 should also have better headings to clearly indicate the variants between panels A-D and E-H, also Figure 4

The cell-cell fusion data in Fig 1C and Supp Figure 3 appear to be contradictory, in that expression of both human and mouse TMPRSS2 in HeLa cells shows quite extensive syncytia (Supp 3), compared to Vero-TMPRSS2 cells (Fig 1C). Often syncytia formation is highly dependent on cell type/growth, as well as expression levels of the protease/receptor, and this should be explained

The authors generally bring a welcome level of nuance to the Omicron entry story, and provide good evidence that TMPRSS2/furin can be used by this virus (likely in a cell/tissue-dependent manner). The data are solid.

Response to Reviewers

REVIEWER COMMENTS

Reviewer #1 (Remarks to the Author):

This submission makes an important observation that helps to clarify some findings in the SARS-CoV-2 / COVID19 field. It shows that the recent SARS-CoV-2 VOCs, particularly omicron, has TMPRSS2 as a host susceptibility factor in laboratory mice. This is important because in vitro (cell culture) results demonstrate that the omicron VOC bypasses the TMPRSS2 activating proteases and instead uses the “late” endosomal proteases for spike-mediated fusion activation. The cell culture results are clearly not able to provide a full picture of the SARS-CoV-2 variants and their utilization of cell surface serine TMPRSS proteases in many infection contexts.

Answer: Thank you for appreciating the significance of this study. As stated in the reviewer's comments, we believe that data from cultured cells alone cannot provide a proper understanding of the characteristics of SARS-CoV-2 variants and that this study using animal models provides essential insights into the knowledge of SARS-CoV-2 variants.

The authors of this submission repeated many of the previously published cell culture findings, and they advanced the cell culture results modestly by including more VOCs in their analysis. This work is solid and convincing. The new findings are with the TMPRSS2 KO mice and this is what gives significance to the report. These findings are valuable for the field, especially because TMPRSS inhibitors such as camostat and nafamostat are promoted as clinically useful antivirals to treat COVID19. However, at present there are some possible caveats to the exclusive use of TMPRSS2 KO mice for analysis. Also, there are some aspects of the manuscript that need to be reconsidered. There are major and minor comments below:

Answer: Thank you for your recognition of our analysis as solid and convincing. We agree with the Editor and Reviewers that we need to be careful about relying solely on knockout (KO) mouse data for our conclusions, and we take their comments seriously. We will discuss the details in the "Response to Comments" section below. Briefly, to overcome the weaknesses of the study using KO mice, we have added two additional experiments using KO and wild-type (WT) mice suggested by the reviewers. One is an analysis using a virus that infects and replicates independently of TMPRSS2, and the other is an analysis using a protease (TMPRSS2) inhibitor and WT mice. Both additional data strengthened our results and arguments, and we thank the reviewers for allowing us to perform these further experiments.

1. It would help if the investigators could challenge WT and KO mice with a

respiratory virus that is known to be completely TMPRSS2-independent. This would provide a control to determine whether the TMPRSS2 KO condition is broadly incapable of supporting a respiratory virus infection (i.e., are there pleiotropic effects of the TMPRSS2 KO condition that render the lung environment less susceptible to many viruses).

Answer: We completely agree with the reviewer. To prove that the reduced infection levels of SARS-CoV-2 in TMPRSS2 KO mice are due to the loss of TMPRSS2 expression and not to other (unknown) factors that these KO mice may have, we performed additional experiments using a virus that infects and replicates independently of TMPRSS2. Specifically, experiments were conducted using the H5N1 subtype influenza virus, which does not require TMPRSS2 for infection and has been shown to grow using another host protease (furin), and the H7N9 subtype, which cannot use furin for growth and requires TMPRSS2. The results showed that, as expected, in the TMPRSS2 KO mice, replication of the H5N1 virus, but not the H7N9 virus, was as efficient as in WT mice. We have added these results **as new data in Supplementary Figure 2** and its contents in the main text (**page 9, line 1-4; page 24, line 2-12**). The same results were also shown in our previous study, which was mentioned in the main text (**page 8, lines 19 - page 9, line 1**).

2. If a TMPRSS2-independent virus is not known or available, then it would help if the investigators evaluated whether nafamostat or camostat suppress the VOC infections in WT mice, down to levels equivalent to those observed in the TMPRSS2 KO mice. This orthogonal pharmacologic approach may be necessary to fully validate the genetic data derived from the TMPRSS2 gene KO condition.

Answer: To strengthen the data obtained with TMPRSS2 KO mice, we conducted an experiment in which WT mice were treated with nafamostat, which inhibits the activity of TMPRSS2. The results showed that, as expected, nafamostat efficiently inhibited the growth of SARS-CoV-2. Furthermore, this inhibitory effect was also observed in the Omicron variant, supporting one of the key messages of our paper that the Omicron variant, like other SARS-CoV-2 strains, utilizes TMPRSS2 for growth in vivo (at least in mice). We have added these results **as new data in Figure 4** and its contents in the main text (**page 12, line 2-11; page 23, line 20 - Page 24, line 2**).

3. Discussion section appears to have biases and some statements do not appear to be adequately supported. For example, lines 249-250, and lines 254-255: IT is really not clear that the VLPCs are “driven by the TMPRSS2 pathway” and it is not clear whether adaptations to the early entry pathway “explain the increase in transmissibility of D614G, alpha and delta. There are many other selective forces at play in determining transmissibility. The discussion section requires several revisions.

Answer: We agree with the reviewer. In the Discussion section, the hypothesis that the use of the furin/TMPRSS2-dependent early entry pathway contributes to human adaptation and human-to-human transmission of SARS-CoV-2 was stated as if it were a proven fact. While certain papers mention such a possibility, it is not a proven fact nor a consensus among researchers, so this idea is one of possibility (a hypothesis). Thus, in the revised manuscript, we stated that under the limited condition that the hypothesis is correct, the increased efficiency of using the early entry pathway could partly explain the high transmissibility of specific variants. **(Page 12, line 18-21)**

4. A key reference is missing; PMID: 25666761. This paper shows that SARS-CoV, a virus that definitely uses the late cathepsin pathway in cell culture, does in fact use the camostat-sensitive TMPRSS pathway in laboratory mice. It should be cited and discussed.

Answer: We appreciate the Reviewer's expert remarks. The paper the Reviewer mentioned is significant in discussing this study's results. This previously published paper (PMID: 25666761) and the one presented here complement each other and emphasize the importance of in vivo experiments (animal studies) in analyzing the availability of TMPRSS2 by coronaviruses (SARS-CoV and SARS-CoV-2). These studies provide a scientific rationale for developing treatments for coronavirus infections using TMPRSS2 inhibitors. The paper was cited and discussed in the revised manuscript. **(Page 14, line 13-20; page 15, line 11-16; reference 39)**

Minor comments:

1. Lines 169-170; “the cathepsin pathway cannot be a substitute for the TMPRSS2 pathway”: This seems to be a significant important conclusion but is not explained or discussed in any way. What are the authors’ speculations about this?

Answer: We appreciate the reviewer's comments on the importance of the conclusions of this study. We have added a brief explanation for why we stated that "the cathepsin pathway cannot be a substitute for the TMPRSS2 pathway". Its contents are "If the cathepsin-mediated endocytosis pathway can also be used for SARS-CoV-2 infection in Calu-3 cells, then a certain level of viral infection should be observed even when TMPRSS2 activity is inhibited by nafamostat. However, in Calu-3 cells, SARS-CoV-2 infection was almost completely inhibited by nafamostat alone". **(Page 8, line 8-14)**

2. Fig 1A; cannot see the SD ranges, and furthermore the P values as noted in the Fig 1 legend are not assigned by asterisks.

Answer: We have revised the figure according to the reviewer's remarks and instructions.

3. Fig 1B; are we looking at duplicate results; why no gamma in the second lower set of results?

Answer: In the manuscript of the original submission, we used a monoclonal antibody (clone E7M5X, Cell Signaling Technology, Danvers, MA, USA, Cat# 42172, 1:1000) to detect the S1 subunit. However, this monoclonal antibody failed to recognize S1 in the Gamma variant, leaving only the Gamma data invisible in the bottom panel of Figure 1B. In the revised manuscript, we tried several monoclonal antibodies to solve this problem and used a monoclonal antibody (clone # 1035206, R&D systems, Cat# MAB105403, 1:500) that could recognize the S1 subunit of all used strains in this study. Therefore, in the revised manuscript, we have included that data. (Page 21, line 8-10; Fig. 1B)

4. Fig 1D; EST or E64d? Are they the same? If the same, then suggest consistence between figure and figure legend.

Answer: Since EST and E64d indicate the same reagent, we followed the reviewer's comment and unified the description to EST.

5. Fig 2B; define “crossing point”; is this the same as “cycle threshold”? (I am not familiar with “crossing point”).

Answer: The Ct (cycle threshold) value is calculated from the intersection of the amplification curve and the threshold value by the crossing point method. It is known that this calculation method is applied in ABI (Thermo) instruments such as 7500Fast, Quantstudio, and Stepone. Ct values may vary even for the same amplification curve, depending on the analysis conditions, such as the threshold value. In contrast, the Cp value is the maximum inflection point calculated by the second derivative maximum method. This calculation method is known to be applied to Roche instruments such as LightCycler 480 and LightCycler 96, etc. Unlike the Ct value, if the amplification conditions are the same, there is not much difference in values from test to test because the threshold value is not used. Indeed, both Ct and Cp are output values of real-time PCR, but they are calculated based on different concepts and should be distinguished. This study used Cp to represent real-time PCR data because the LightCycler instrument was used.

6. Figs 3 and 4 are essentially one figure.

Answer: Following the Reviewer's comment and Nat Communications' instructions

for formatting figures we have created a figure integrating Fig. 3 and Fig. 4.

Reviewer #2 (Remarks to the Author):

This is a timely article that addresses an important topic – the activation of SARS-CoV-2 Omicron spike and the entry of the virus into cells. The authors portray a slightly different story to the prevailing view that Omicron has a very different entry pathway to other variants. The current work suggests a shift in pathway, but no fundamental change in the activating protease (TMPRSS2).

Answer: Thank you for recognizing our manuscript as a timely one on an important topic. We also appreciate the Reviewer's essential and helpful comments. We have taken all comments seriously and have made revisions.

A general comment is that SARS-CoV-2 entry is complex and variable and it is easy to draw inappropriate conclusions. With this in mind I would suggest the authors be more careful with wording in the title and abstract – eg saying that multiple entry pathways exist, rather than there two.....” – recent studies have indicated (rather than ‘shown’). What is meant by “improved’ use (line 35), also “not impaired” (line 36). They should specify cell types in the abstract (line 37). It is also worth considering the use of “in vivo” in the Title – as this really refers to mice

Answer: Thank you for the Reviewer's valuable and essential comments. We agree with the Reviewer's comments. Indeed, the cell entry mode SARS-CoV-2 is complex and cannot be explained by at least the two pathways mentioned in this paper. Although these two pathways are significant, many others exist. Also, the results presented in this paper are from a mouse model. Therefore, to avoid any misleading, we have made it clear throughout the manuscript, including in the revised title, that the data are from analyses of mice. In addition, we omitted inappropriate descriptions because the number of words in the abstract had to be modified from the original 200 to 150 according to Nat Communications' instructions. We also changed the ambiguous description "not impaired" to "efficiently" to make it easier to understand. (Page 2, line 2-15)

The manuscript would be improved by better annotation of the figures (headings for cell types etc).

Answer: Thank you for the Reviewer's comments. We have modified the Figures as per the Reviewer's suggestion.

Figure 1B is very unclear – it is not labelled well, and there seems to be a missing lane for the lower panel (was gamma not tested?). The bar charts in panel C are very hard to visualize, and statistics needs to be clearer (what is being compared with the **). It seems that Omicron spike is less well incorporated into particles (is this true?) but cleaved once it is present?

Answer: Thank you very much. Indeed, the labeling in Fig. 1B was inadequate. It was also difficult to distinguish individual symbols in other figures, and statistical comparisons were unclear. All of these points have been improved. Also, the Gamma variant data were not detected due to a problem with the antibody, so we changed the antibody and repeated the experiment. In the revised version, we included data showing the Gamma variant and other variant signals. (**Page 21, line 7-8; Fig. 1B**)

Some previous papers showed that the S protein of the Omicron variant was not efficiently incorporated into the particles. To investigate this point, we detected N protein in addition to S protein. We added that data to Fig. 1B. This data alone cannot be used to evaluate the incorporation of S proteins into particles. Still, the data do not suggest that incorporating S proteins into particles is inefficient. (**Page 21, line 11-12; Fig. 1B**)

It has been reported in many papers that the S protein of the Omicron variant is not easily cleaved. Therefore, we analyzed that point very carefully in this paper. However, none of the data supported that the S protein of the Omicron variant is poorly cleaved, but rather that it is efficiently cleaved. Other studies have also supported the efficient cleavage of the Omicron S proteins, and our results support these findings.

Figure 3 should also have better headings to clearly indicate the variants between panels A-D and E-H, also Figure 4

Answer: Thank you for the Reviewer's comments. We have modified the Figures as per the Reviewer's suggestion. Also, following Reviewer1's comment and Nat Communications' instructions for formatting figures, we have created a figure integrating Fig. 3 and Fig. 4.

The cell-cell fusion data in Fig 1C and Supp Figure 3 appear to be contradictory, in that expression of both human and mouse TMPRSS2 in HeLa cells shows quite extensive syncytia (Supp 3), compared to Vero-TMPRSS2 cells (Fig 1C). Often syncytia formation is highly dependent on cell type/growth, as well as expression levels of the protease/receptor, and this should be explained

Answer: We thank the Reviewer for his expert and thoughtful review. As the

Reviewer points out, the level of syncytium formation by viral membrane fusion proteins is greatly influenced by various factors, including the cells used, receptor expression levels, protease levels, cell density, etc. In the experiment in Figure 1C, VeroE6/TMPRSS2 cells, a cell line that constitutively expresses ACE2 and human TMPRSS2, were used. In Supplementary Figure 3, HeLa cells constitutively express human TMPRSS2 or mouse TMPRSS2 (HeLa/hTMPRSS2 cells and HeLa/mTMPRSS2 cells, respectively) were used. In Supplementary Figure 3, HeLa cells were used instead of VeroE6/TMPRSS2 cells because cells that constitutively express mouse TMPRSS2 were needed. Greater levels of syncytium-formation were observed in HeLa cells (HeLa/hTMPRSS2 cells or HeLa/mTMPRSS2 cells) than in VeroE6/TMPRSS2 cells when the S protein of SARS-CoV-2 (including Omicron variant) was expressed. Its underlying mechanism was unclear. However, since the purpose of the analysis in Supplementary Fig. 3 is to show that mouse TMPRSS2 can activate the S protein of the Omicron variant as efficiently as human TMPRSS2, we believe that the purpose of the experiment was achieved. We have added explanations in the text to clarify that different cells are used in Figure 1C and Supplementary Figure 3. **(Page 11, lines 4-9)**

The authors generally bring a welcome level of nuance to the Omicron entry story, and provide good evidence that TMPRSS2/furin can be used by this virus (likely in a cell/tissue-dependent manner). The data are solid.

Answer: Thank you for appreciating that our research paper provides good evidence and shows solid results.

REVIEWERS' COMMENTS

Reviewer #1 (Remarks to the Author):

This revised submission has taken reviewer comments seriously and has made several valuable additions. The report has increased impact. It will be important to the field in that the TMPRSS proteases will now be viewed as in vivo host susceptibility factors for all SARS-CoV-2 VOCs including omicron. This resets the field appropriately on host determinants of virus susceptibility.

REVIEWERS' COMMENTS

Reviewer #1 (Remarks to the Author):

This revised submission has taken reviewer comments seriously and has made several valuable additions. The report has increased impact. It will be important to the field in that the TMPRSS proteases will now be viewed as in vivo host susceptibility factors for all SARS-CoV-2 VOCs including omicron. This resets the field appropriately on host determinants of virus susceptibility.

Authors' response

We thank the reviewers for their careful review, hard work, and valuable comments.